# Phanerozoic Evolution of Continental Large Igneous Provinces: Implications for Galactic Seasonality

**Victor P. Nechaev [1,*], Frederick L. Sutherland [2] and Eugenia V. Nechaeva [3]**

[1]  Far East Geological Institute, Far Eastern Branch, Russian Academy of Sciences, 159, Prospect 100-letiya, 690022 Vladivostok, Russia
[2]  Geoscience, Australian Museum, 1 William Street, Sydney, NSW 2010, Australia
[3]  Polytechnic Institute, Far Eastern Federal University, 690922 Vladivostok, Russia
[*]  Correspondence: nechaev@fegi.ru; Tel.: +7-902-4866273

**Abstract:** This study reviews the available data on the Phanerozoic plume activity (Large Igneous Provinces (LIP's) size and frequency) and geochemistry of their igneous rocks. A major goal of this review is to try to find the changes in intensity and geochemistry of mantle plumes linked to the Earth's evolution and galactic seasonality that was supposed in the authors' previous publications. The data indicate that the Cambrian–Ordovician and Jurassic–Cretaceous galactic summers were associated with peaks of various igneous activities including both plume- and subduction/collision-related magmatism, while the Carboniferous–Permian and current galactic winters led to significant drops within the igneous activity. The materials subducted into the transitional and lower mantle, which highly influenced the plume magmas in the galactic-summer times, were less significant in the galactic spring and autumn seasons. The least subduction-influenced LIPs were probably the Tarim and Emeishan deep plume magmas that developed in the mid–late Permian, during the galactic late winter–early spring subseason. The Fe enrichment of clinopyroxenite, gabbro, and associated ores of these provinces might be caused by fluids ascending from the core–mantle boundary. However, the most significant core influence through plume-associated fluids on the surface of solid Earth is supposed to have occurred in the galactic summer times (Cambrian–Ordovician and Jurassic–Cretaceous), which is indicated by peak abundances of ironstone ores. Their contributions to the Cambrian–Ordovician and Jurassic–Cretaceous plume magmas were, however, obscured by more significant influences from subduction.

**Keywords:** mantle plume; subduction; crustal contamination; core–mantle interaction; geological evolution

## 1. Introduction

A major goal of this review is to try to find the evolutionary changes in intensity and geochemistry of mantle plumes linked to the Earth's evolution and galactic seasonality that was supposed in our previous publications [1,2]. The earlier paper [1] suggested that the galactic seasons of the Earth indicate significant changes caused by its distance from the Sun while that star was flying along its elliptical orbit. Under the gravitational influence of a huge mass at the galactic center, the Solar System, including Earth, became extended when it moved closer to the center and then contracted back towards the Sun when it became more distant. So, the galactic winters on the Earth coincided with closer to the galactic center position of the Solar System and vice versa. Galactic winters occurred on Earth during the Vendian, Carboniferous–Permian and Late Cenozoic periods. These times were characterized by long-term decreases in global temperature and biodiversity, and by formation of supercontinents. In the warmer galactic seasons, the Earth would have some semblance of Venus conditions with its widespread mafic volcanism, disseminated thicker crust, and dense 'gas-laden' atmosphere [3]. It was suggested that

an extensive basic magmatism grew the thinner oceanic-type crust and developed large igneous provinces (plume tectonics?) during the spring–summer seasons. Intense folding, dynamic metamorphism and acidic (subduction/collision-related) magmatism grew the thicker continental-type crust during the autumn–winter seasons. The later work devoted to the Cretaceous turn of geological evolution [2], however, showed that the galactic summers were associated with peaks of varied igneous activities. These included both plume- and subduction/collision-related magmatism, while the galactic winters showed significant drops of any igneous activity. The more recent works [4,5] demonstrated a close relationship between plume- and subduction/collision-related activities in Northeast Asia for the Mesozoic time and thus supported the latter supposition.

Below, we will test our previous ideas by reviewing the Phanerozoic volcanic activity of Large Igneous Provinces (LIPs), especially focusing on the largest and best-studied provinces emplaced on continents. The continental LIPs were selected because the alternative oceanic LIPs are poorly preserved in the pre-Jurassic geological records. The siliceous LIPs are relatively rare phenomena that are not suitable for wide comparison. They were also deeply influenced by crustal contamination that does not contribute to the achievement of this study goal.

The selected Paleozoic LIPs [6] include: Kalkarindji (about 510 Ma) from north Australia; Yakutsk-Vilyui (about 370 Ma) and Siberian Traps (about 252 Ma) from north Siberia; Kola-Dnieper (about 370 Ma) from Eastern Europe; Skagerrak-centered (about 300 Ma) from Scandinavia; Tarim (about 285 Ma) from Central Asia; and Emeishan (about 260 Ma) from South China. The Mesozoic continental LIPs are more extensive. Particularly, the Central Atlantic Magmatic Province (about 201 Ma) partly cover North America, Europe, Africa, and South America. The Karoo (South Africa) and the Parana (eastern South America)—Etendeka (western South Africa) provinces were emplaced in Gondwana at about 183 Ma and 134 Ma, respectively. Fragments of the High Arctic LIP (about 121 Ma) were found in North America, Asia, and Europe [6]. Fragments of the Deccan LIP (66 Ma) remain in India, as well as on the Chagos–Laccadive Plateau and Seychelles. The less abundant Cenozoic LIPs include the North Atlantic Igneous Province (60 Ma) in Europe and Greenland, Afro-Arabian LIP (31 Ma) in East Africa and Arabian Peninsula and Yellowstone–Columbia River LIP (17-0 Ma) in North America.

## 2. Data Sources and Methods of Interpretation

The geochemical data for this review were downloaded from the GEOROC database [7] in April–May 2022, using the following parameters: geological settings = Continental Flood Basalt + Intracontinental Volcanics + Rift Volcanics and LIP's name, with additions from the following research papers: [8] for the Kalkarindji (Australian craton) LIP; [9,10] for the Yakutsk–Vilyui (North Siberia) LIP; [11–14] for the Kola–Dnieper (East European craton) LIP; and [15] for the Karoo LIP. The data interpretation was conducted using diagrams and comments on them from [16–18]. Names, ages and sizes of Phanerozoic LIPs are after [6].

## 3. Results and Discussion

### 3.1. LIP's Size (Plume Activity) in the Geological Record

Figure 1E shows how the LIP's size and frequency changed in Phanerozoic in comparison with other igneous activities (Figure 1A–D). The former events tend to decrease unevenly with age, although they are not reliably evaluated for every province [6]. Thus, these decreases may be the quasi decreases caused by poorer preservation of the older provinces.

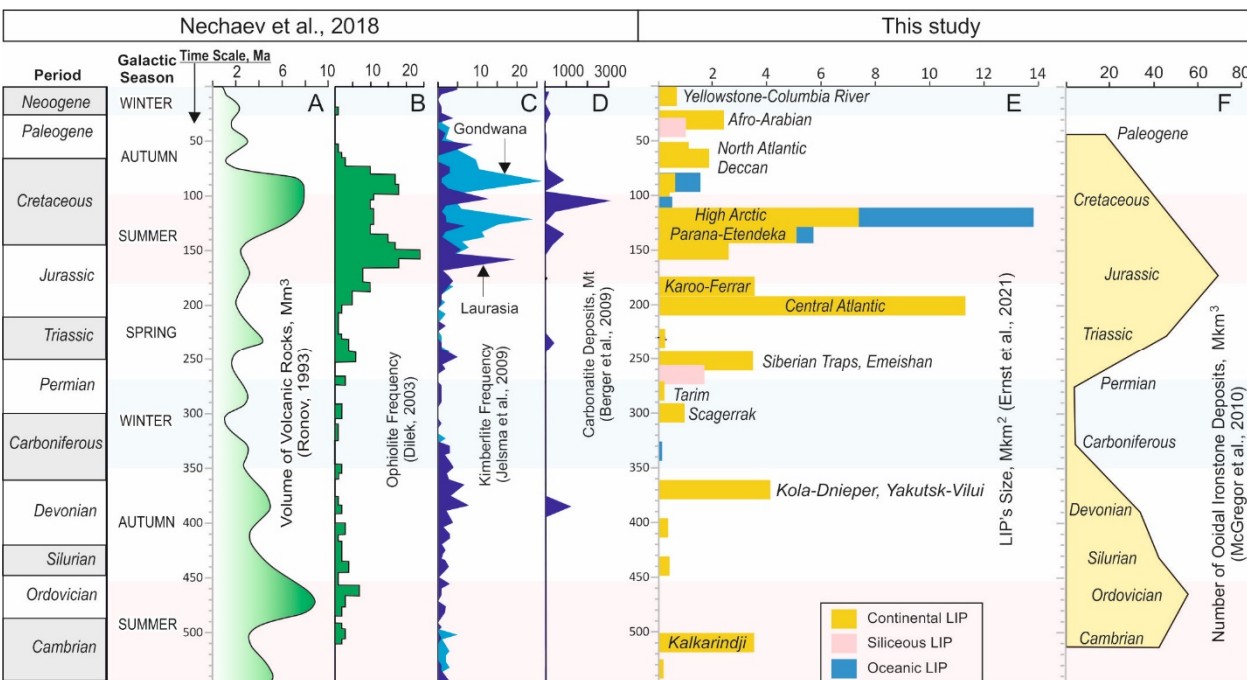

**Figure 1.** The proposed galactic seasons (after [2] in comparison with global changes in igneous and iron ore-forming activities (after [19–21]) with additions from [6,22]). (**A**) volume of volcanic rocks; (**B**) ophiolite frequency; (**C**) kimberlite frequency; (**D**) carbonatite deposits (**E**) Lip's size (**F**) number of ooidal ironstone deposits. The LIPs names are shown only for the provinces for which geochemical data are used in this study.

It is also well seen in Figure 1E that the LIP's activity in the Cambrian–Ordovician, and especially in the Jurassic-Cretaceous galactic summers, was much more extensive than in the Carboniferous–Permian and recent galactic winters. This strongly supports our recent suggestion that the galactic summers were associated with peak plume activity, while the galactic winters led to its significant drops [2].

### 3.2. Plume–Mantle and Plume–Subduction Interactions

Zhang et al. [18] considered LIP's compositions related to plume interactions with the core–mantle boundary and the transitional mantle zone. As a result, three major plume types were geochemically distinguished, including:

1.  Andersonian plumes that originated from the upper mantle above the transitional mantle zone, characterized by the basalt–rhyolite (BR) rock suite;
2.  Morganian plumes that derived from the core–mantle boundary but not significantly reacted with the transitional and upper mantle, characterized by both BR and basalt–trachyte (BT) suites;
3.  Super plumes derived from the core–mantle boundary and widely reacted with the transitional and upper mantle, characterized by a complex rock assemblage including BR, BT, and basalt–phonolite (BP) suites.

The Super and Morganian plumes are suggested to be sourced by the lower mantle and subducted materials that had reached the lower mantle and core–mantle boundary as a result of mantle avalanches [23]. It is also supposed that the liquid outer core is the source of Fe for the bridgmanite and post-perovskite phase transition in the lower mantle [18].

The TAS diagrams show that the winter LIPs (Skagerrak, Tarim, and Yellowstone; Figure 2C,E) are of the BR suite indicating the Andersonian plumes, although the Tarim suite includes phonolitic rocks reflecting some minor influence of the lower mantle/core

source. This confirms our prediction that the galactic winter times are associated with a relatively low plume activity.

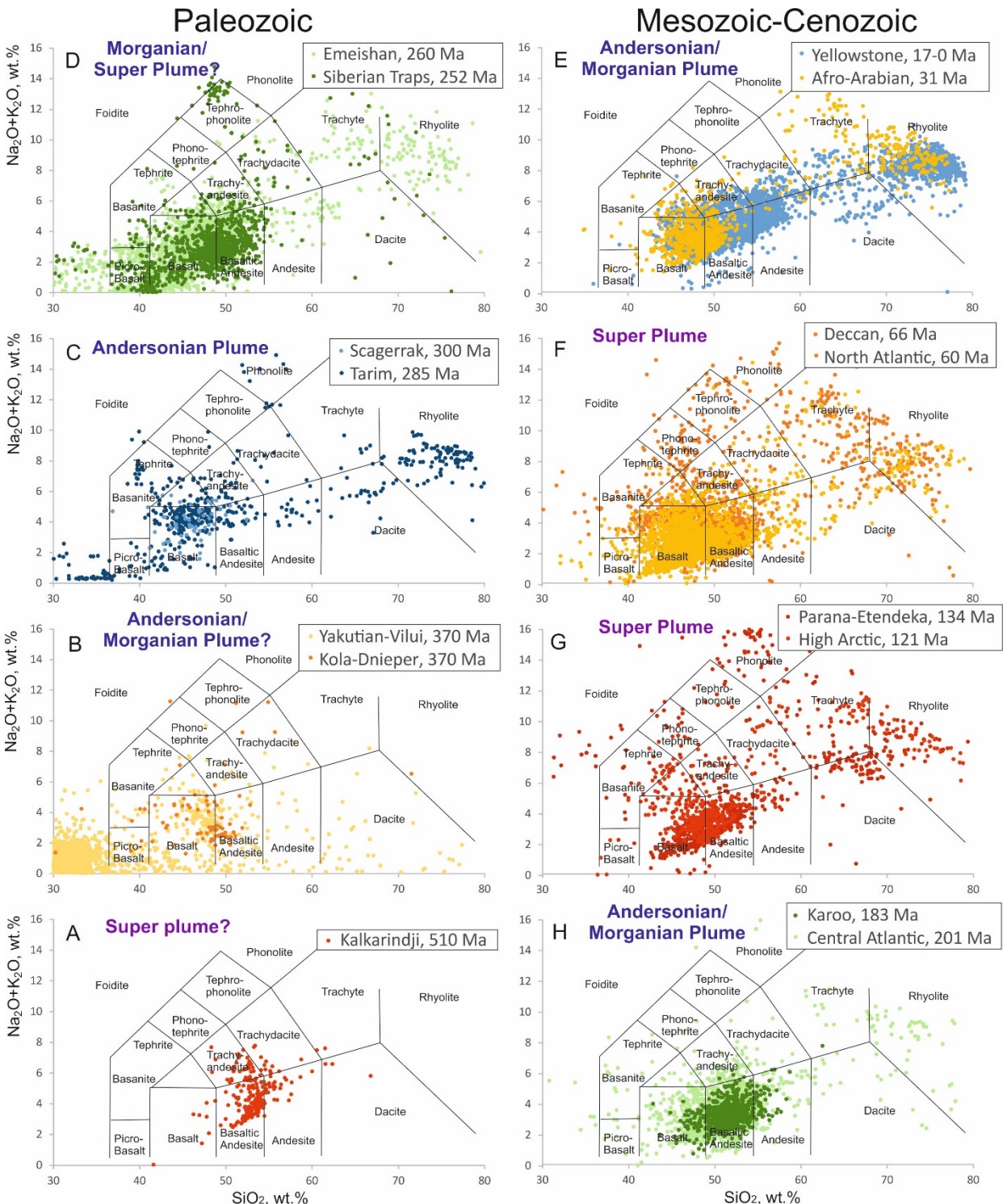

**Figure 2.** TAS diagrams for distinguishing the selected LIPs between the Andersonian, Morganian and Super plumes according to the discrimination by Zang et al. [18]. Plots of the LIPs attributed to the galactic spring, summer, autumn, and winter times (see Figure 1) are colored in green, red, yellow–orange, and blue, respectively. Data sources are cited in Section 2. (**A**) Kalkarindji; (**B**) Yakutian-Vilui and Kola-Dnieper; (**C**) Scagerrak and Tarim; (**D**) Emeishan and Siberian Trips; (**E**)

Yellowstone and Afro-Arabian; (**F**) Geccan and North Atlantic; (**G**) Parana-Etendeka and High Arctic; (**H**) Karoo and Central Altantic.

In contrast, the summer LIPs (Parana–Etendeka and High Arctic; Figure 2G), which are characterized by the BP-BT-BR suites, may be confidently attributed to the Super plumes. This confirms our prediction that the galactic summer times are associated with a peak plume activity.

The North Atlantic, and especially the Deccan LIPs, are identified as the Super plumes that may be caused by a delayed galactic-summer effect [2]. Of note, however, is that the Kalkarindji LIP, which formed in the older galactic-summer time, may not be fully identified within the available data (Figure 2A).

The other provinces under consideration belong to the transitional Andersonian–Morganian plumes that are undeveloped in comparison with the Super plumes. This is also in good agreement with the hypothesis of galactic influence on the Earth's evolution [1,2].

A correlation between plume activity and plume depth may not be direct. However, a higher turbulence in the liquid core during the galactic summers could result in additional activity of the Morganian and Super plumes. The Andersonian plumes might also be more active due to more dynamic asthenospheric flows. A higher plume activity generated more plume rocks that came from different source regions.

The diagrams of Figure 3 are used to monitor subduction-metasomatism and crustal assimilation (Th/Nb, a crustal input proxy) and depth and degree of mantle melting (Ti/Yb, a residual garnet proxy) [16,17]. They distinctly show that all the tested LIPs were derived from a deep, hot and enriched mantle source. At the same time, the galactic-winter Andersonian-type LIPs (Skagerrak, Tarim, and Yellowstone; Figure 3C,E) and some galactic-autumn Andersonian/Morganian-type LIPs (Yakutian–Vilyui and Kola–Dnieper) almost devoid of influence of subduction, while the galactic-spring and, especially, galactic-summer LIPs have a significant subduction contribution. This confirms the linkage between the igneous activity and the galactic seasonality that was predicted by the hypothesis of galactic influence on the Earth's evolution [2].

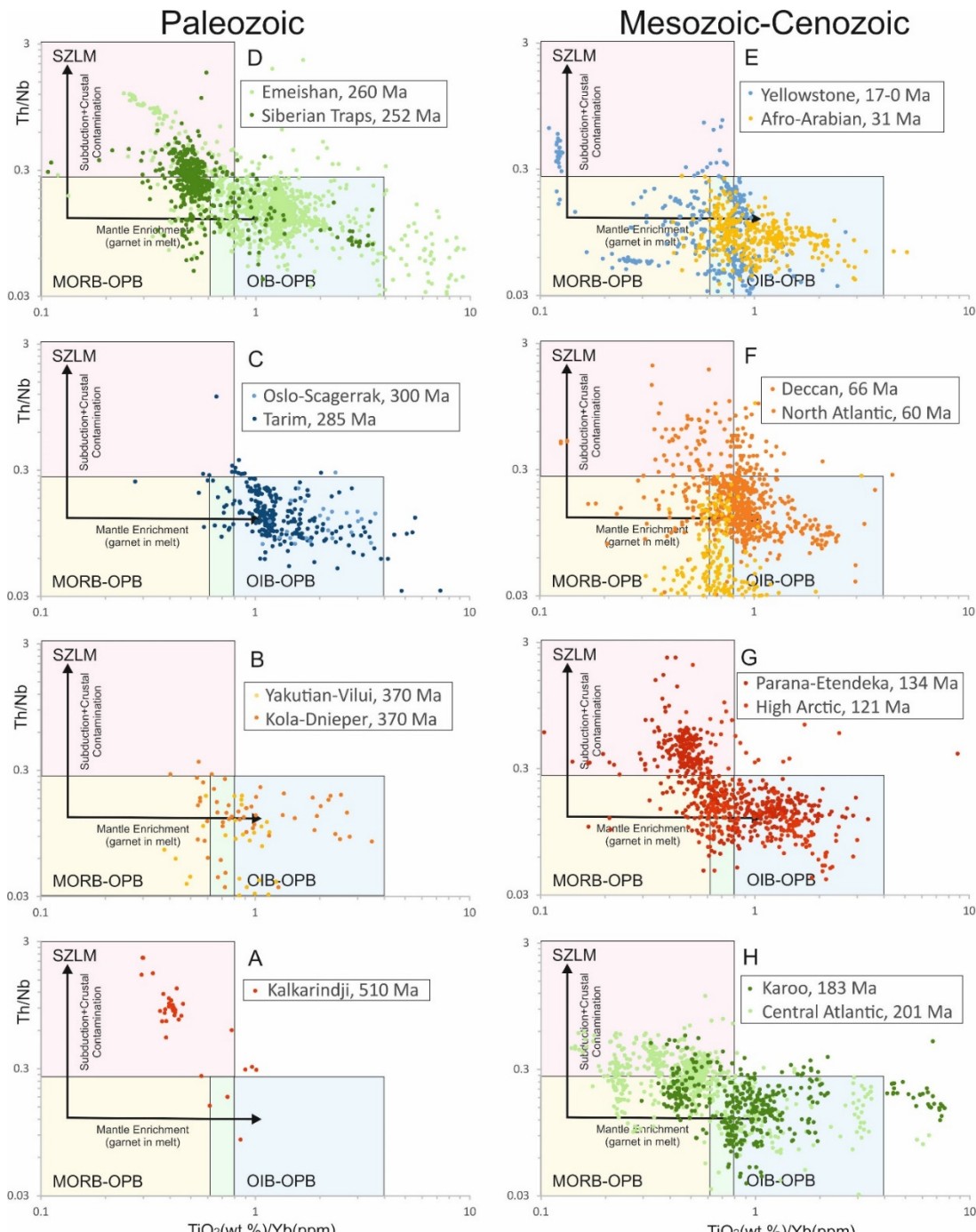

**Figure 3.** Th/Nb vs. TiO₂/Yb in mafic rocks (SiO₂ = 41–52 wt.%) from the selected LIPs on the LIP printing diagram with a subduction-modified lithospheric mantle (SZLM) array and a 'plume' array [16,17]. MORB, mid-oceanic ridge basalt; OIB, oceanic island basalt; OPB, oceanic plateau basalt. Data sources are cited in Section 2. (**A**) Kalkarindji; (**B**) Yakutian-Vilui and Kola-Dnieper; (**C**) Scagerrak and Tarim; (**D**) Emeishan and Siberian Trips; (**E**) Yellowstone and Afro-Arabian; (**F**) Geccan and North Atlantic; (**G**) Parana-Etendeka and High Arctic; (**H**) Karoo and Central Altantic.

### 3.3. Plume–Core Interaction

The hypothesis of galactic influence on the Earth's evolution [1,2] states that a liquid nature of the Earth's core may have reacted to the gravitational and electromagnetic transformations in the outer space. In particular, intensive turbulent flows might happen in the outer core during the galactic summer time. This would have favored the rise of

voluminous magmatic plumes and associated fluid flows. It is reasonable to suggest by contrast that the winter times would have been associated with the more stable core and a decrease in core–plume interaction. These ideas may be developed as follows.

The Earth's core principally consists of iron [24], whereas the mantle, where plumes are born, is predominated by silicates. Thus, an increased plume–core interaction in the galactic summer times might lead to an iron contribution from the core into the silicate mantle melts that should be indicated by higher iron/silica ratio in the plume-generated rocks. Figure 4 testifies to this supposition.

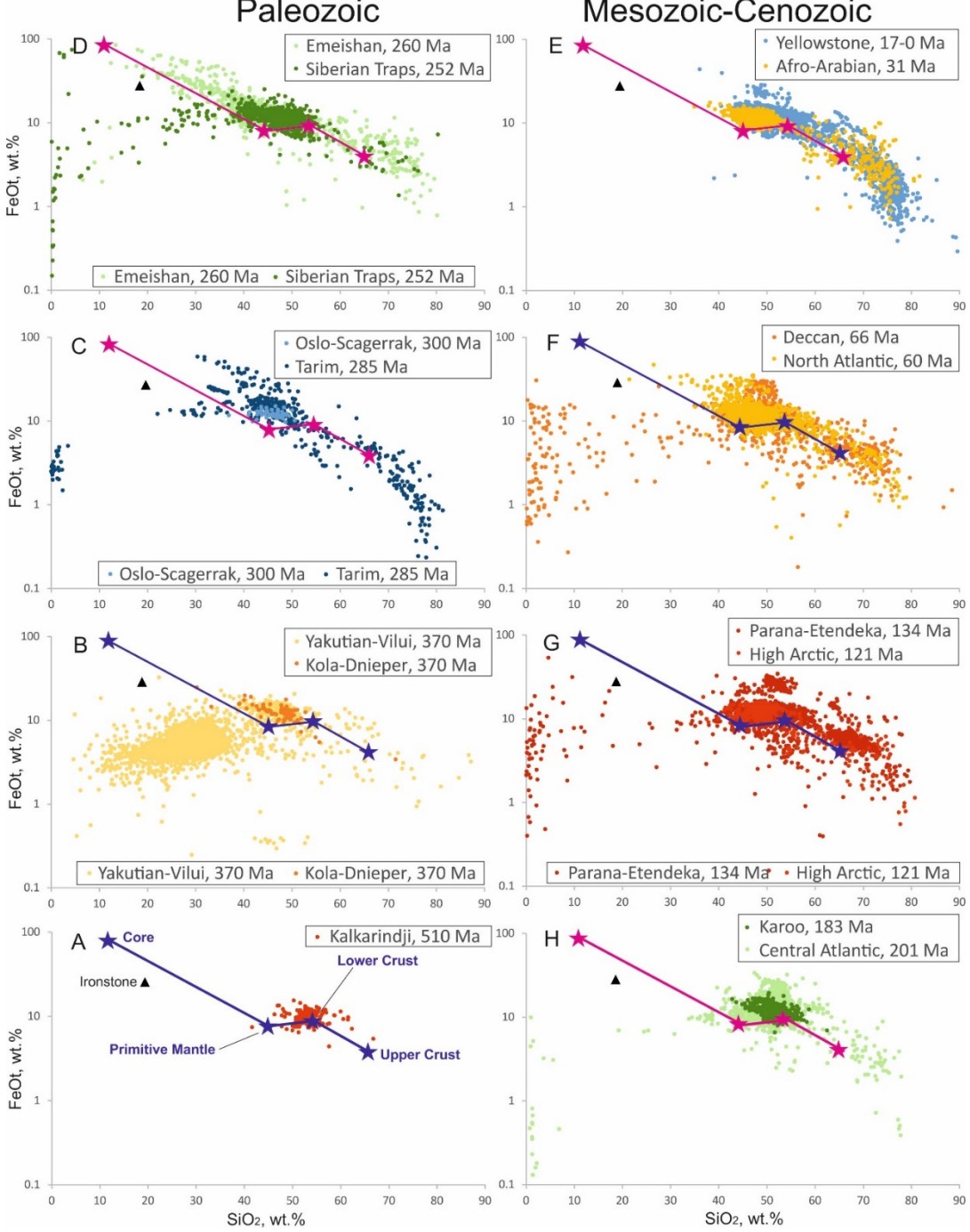

**Figure 4.** The $SiO_2/FeO_t$ ratio in magmatic rocks of the selected LIPs. $FeO_t$ is total iron as FeO. Compositions of ironstone deposits, the core, primitive mantle, and crust are shown after [22,24–26], respectively. Data sources are cited in Section 2. (**A**) Kalkarindji; (**B**) Yakutian-Vilui and Kola-

Dnieper; (**C**) Scagerrak and Tarim; (**D**) Emeishan and Siberian Trips; (**E**) Yellowstone and Afro-Arabian; (**F**) Geccan and North Atlantic; (**G**) Parana-Etendeka and High Arctic; (**H**) Karoo and Central Altantic.

The diagrams provide evidence that iron content in mafic, intermediate and felsic rocks of all the tested LIPs is higher than in the average compositions of mantle and crust, while that of the ultramafic and carbonatite rocks ($SiO_2 < 41$ wt.%) is mainly lower than in the primitive mantle–core trend. This may be explained by a higher fusibility (lower melting point) of ferriferous minerals in comparison with the high-magnesian ones during partial melting followed by magma fractionation in the mantle and under crustal conditions [27,28]. This seems a normal trend for the partial melting and FC process of a large degree melt system. Tholeiitic magma systems commonly show this trend. However, the magma melting and fractionation may obscure a primary iron enrichment related to the plume–core interaction. This suggestion is supported by exclusive compositions of clinopyroxenite and gabbro intrusions from the Tarim and Emeishan LIPs [29,30], which are characterized by the $FeOt/SiO_2$ plots positioned above and along the primitive mantle–core trend (Figure 4C,D). These rocks and associated Fe–Ti ores were fractionated from "a Fe highly enriched parental magma resulted from partial melting of a metasomatized lithospheric mantle" [29]. If so, the Fe enrichment of parental magma might form under the influence of fluids derived from a plume–core interaction that we suggested above. It is important for this study that the Fe-bearing Tarim and Emeishan LIPs are dated for the relatively short late winter—early spring galactic season (285–260 Ma; Figure 2C,D). We may suppose from the latter and our findings described in Section 3.2 that plume–subduction interaction in this galactic subseason remained rather low that allowed the advancing core influence on the deep plume roots to express itself most explicitly. At the same time, the upper parts of these plumes might be influenced in different degree by the crustal materials (Figure 3C,D).

To provide further comparative background on the Tarim and Emeishan plume provinces, some additional studies are cited here to outline their characteristics. The older Tarim province, derived from a long-term, enriched continental lithospheric mantle domain, lacks Phanerozoic metasomatic influences induced by subducted materials, and includes a plume incubation stage [31–33]. The younger Emeishan province also suggests a mantle plume role [34], though of uncertain nature. Experiments, however, supported a robust plume presence [35]. In addition, many studies evidence that the Emeishan and Tarim LIPs including Fe–Ti ores have been affected by subduction [36–45]. This seems contradictory to our suggestion that the Tarim and Emeishan plumes were the least subduction-influenced. However, we do not completely deny the effect of subducted materials on the Tarim and Emeishan magmas. Moreover, the above-referenced studies did not define a temporal correlation between the subduction and plume processes. For example, Hou et al. [36], who revealed low $^3He/^4He$ and $^{40}Ar/^{36}Ar$ ratios of olivine and clinopyroxene grains separated from the Fe–Ti–V oxide ore-bearing intrusions in the Emeishan LIP, concluded that they are due to "subduction-related fluids and melts that had been stored in the lithospheric mantle for long periods. Considering the regional geologic history, such addition can be attributed to the paleo subduction that occurred along the western margin of the Yangtze Block during the Neoproterozoic." Thus, the mentioned studies do not contradict our suggestion that the Tarim and Emeishan deep plume magmas were the least influenced by contemporary subduction.

The time distribution of ironstone deposits (Figure 1F [22]), which were probably sourced by plume-related hydrothermal fluxes [46], supports our suggestion of iron enrichment originating from the plume–core interaction. Indeed, the highest peaks of ironstone formation and plume activity occurred in the Jurassic–Cretaceous galactic-summer time. A less significant Cambrian–Ordovician increase in ironstone formation also took place in the galactic-summer time, which was associated with the high level of

global volcanic activity [19] and the extensive (3.51 Mkm$^2$ [6]) Kalkarindji plume (Figure 1A,E,F). In addition, the ironstone ores are close in composition (FeOt/SiO$_2$) to the Earth's core–primitive mantle trend (Figure 2). In contrast, the lowest level of ironstone formation is recorded in Carboniferous and Permian that is during the galactic winter ([22] Figure 1F).

### 4. Conclusions

The available data in this study indicate that the Phanerozoic plume activity including the LIP's frequency and size were significantly higher in the Cambrian–Ordovician and, more distinctly, Jurassic–Cretaceous galactic-summer times. However, this study has not supported our primary suggestion [1] that the plume-related magmatism and tectonics were more active during the galactic spring–summer seasons, while the subduction/collision-related magmatism and tectonics were more intense during the autumn–winter ones. Actually, the galactic summers were associated with peaks of various igneous activities including both plume- and subduction/collision-related magmatism, while the galactic winters led to significant drops of any igneous activity. What is more, the materials subducted into the transitional and lower mantle highly influenced the plume magmas in the galactic-summer times and less significantly in the galactic spring and autumn seasons. The least subduction-influenced probably were the Tarim and Emeishan deep plume magmas that occurred in the mid–late Permian, during the galactic late winter–early spring subseason. The Fe enrichment of clinopyroxenite, gabbro and associated Fe–Ti ores of these provinces might be, at least partly, caused by fluids ascending from the core–mantle boundary. However, the most significant core influence through plumes on the surface of solid Earth is supposed to have occurred in the galactic summer times (Cambrian–Ordovician and Jurassic–Cretaceous), which is indicated by peak abundances of ironstone ores. Their contributions to the plume magmas were, however, obscured by more significant influences from subduction.

**Author Contributions:** V.P.N., Conceptualization, methodology, writing—original draft preparation; F.L.S., writing—review and editing; E.V.N., visualization. All authors have read and agreed to the published version of the manuscript.

**Funding:** This research was supported by regular resources available to Far East Geological Institute, Far Eastern Branch, Russian Academy of Sciences and Far Eastern Federal University. It did not receive any specific grant from funding agencies in the public, commercial, or not-for-profit sectors.

**Data Availability Statement:** Most of the data supporting reported results can be found at the freely available GEOROC database [(https://georoc.eu/)] last assessed on 15 May 2022. Other data were taken from the literature referenced in Section 2.

**Acknowledgments:** The authors highly appreciate the discount of processing charge offered by MINERALS for this article. We also are grateful to Richard Ernst from Carleton University, Ottawa, Canada and three anonymous reviewers for their useful comments on the manuscript.

**Conflicts of Interest:** The authors declare no conflict of interest.

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
