# Peer review of "Phanerozoic Evolution of Continental Large Igneous Provinces: Implications for Galactic Seasonality"

_minerals, doi:10.3390/min12091150_

Round 1

Reviewer 1 Report

This is an interesting work and a mind-opening presentation. Authors used the scale of galaxy to link the Earth's core-mantle activities with the position of solar system in galaxy. Large igneous provinces, their size, geochemical characteristics, and developing time haven been used to support the idea that the position of solar system in galaxy affects the Earth's core-mantle activities. The argument is fascinating and this galactic scale might affected the earth system, since Rodinia breaking up around the time of previous galactic summer. The evidences from this presentation can be explained by other geological activities, though.

Author Response

This is an interesting work and a mind-opening presentation. Authors used the scale of galaxy to link the Earth's core-mantle activities with the position of solar system in galaxy. Large igneous provinces, their size, geochemical characteristics, and developing time haven been used to support the idea that the position of solar system in galaxy affects the Earth's core-mantle activities. The argument is fascinating and this galactic scale might affected the earth system, since Rodinia breaking up around the time of previous galactic summer. The evidences from this presentation can be explained by other geological activities, though.

General Response: Thanks for this evaluation. For sure, our ideas form a hypothesis that needs more studies to become a theory. However, it would be difficult to find some intraterrestrial (geological) explanations of the cyclical global changes spanning hundreds of million years. Thanks also for your deep and meaningful comments in the pdf file.

Responses to the comments in the pdf file:

Last paragraph on Page 3. “In 2 and 3, those plumes derived from core-mantle boundary. How can they sources from subducted plate?

Response: Subducted materials can submerge into the lower mantle and the core-mantle boundary as a result of mantle avalanches (Muller, 2002; https://doi.org/10.1029/2002GL015938). The corresponding addition to the main text and References has been made.

Last paragraph on Page 4 and the next paragraph on Page 5. “Does plume activity have direct correlation with plume depth? If so. what cause that?” “Why plume activity is correlated with depth? Is it possible that high plume activity generated more plume rocks and they came from different source region and then generate different varieties of rocks?”

Response: A correlation between plume activity and plume depth may not be direct. However, a higher turbulence in the liquid core during the galactic summers could result in additional activity of the Morganian and Super plumes. The Andersonian plumes might also be more active due to more dynamic asthenospheric flows. A higher plume activity generated more plume rocks that came from different source regions. The above sentences were added to the text before Fig. 3.

The paragraph at the end of Page 6 and beginning of Page 7. “The garnet zone has a wide depth, how the degree of melting change this ratio? And how the Ti/Yb ratio give the depth information? More details is helpful here” and “how to tell these based on the plots? Needs more explanation”.

Response: More explanations may be found in the referenced works by J. Pearce [16.17]. We avoid to discuss many details in this review to present our general ideas clearer and easier to understand for a wide range of readers.

The first paragraph of Section 3.3, Page 7. “This is an interesting claim, is there any evidence how much in what kind of degree the gravitational change could be? We do see sea level change when the moon is in different position. Is there any kind of information about the galactic position on the surface of the Earth?”

Response: Right, a degree of the gravitational change on the surface of the Earth with respect to our position in the galactic orbit is an important issue.  Unfortunately, it has not been calculated yet. However, the interested readers may learn more about a correlation between sea level changes and galactic seasonality in our previous publications [1,2] that are freely available online.

 The last paragraph on Page 8. “This seems a normal trend for the partial melting and FC process of a large degree melt system. Tholeitic magma system shows this trend”.

Response: You are right, as always. The corresponding addition has been added.

 The paragraph before Section 4. “Ooids formed on the sedimentary condition and near surface hydrothermal events can be correlated with that, why it has to be plume-related?”

Response: The required explanations may be found in the referenced work by A. Bekker et al. [36]. We avoid to discuss many details in this review to present our general ideas clearer and easier to understand for a wide range of readers.

Reviewer 2 Report

The manuscript entitled ‘Phanerozoic evolution of continental large igneous provinces: Implications for galactic seasonality’ by Nechaev et al., reviews the available data from the global Phanerozoic plume activities and geochemistry of their igneous rocks to find the evolutionary changes in intensity and geochemistry of mantle plumes. The topic is good for current climatic change. Although I am not an expert of the continental large igneous provinces and climatology, I try to do a careful reading the manuscript. I feel that the discussion section is a little too simple and suggest that authors should further discuss on the relationship between galactic seasonality and continental large igneous provinces in details. Besides, many grammatical problems exist in the text and language should be polished thoroughly by a native speaker. The followings are my comments on this manuscript. 

Page 1: ‘The materials subducted into the transitional and lower mantle highly influenced the plume mag[1]mas in the galactic-summer times and less significantly in the galactic spring and autumn’ should be ‘The materials subducted into the transitional and lower mantle, which highly influenced the plume mag[1]mas in the galactic-summer times and less significantly in the galactic spring and autumn’.

Page 1: ‘The least subduction-influenced probably were the Tarim and Emeishan deep plume magmas…’ should be ‘The least subduction-influenced probably was the Tarim and Emeishan deep plume magmas…’.

Page 1: ‘…, that is indicated by peak abundances of ironstone ores’ should be ‘…, which is indicated by peak abundances of ironstone ores’.

Page 1: ‘while that star was flying along its elliptical orbit’ should be ‘while stars were flying along its elliptical orbit’.

Page 1: Galactic winters occurred during the Vendian, Carboniferous-Permian and Late Cenozoic periods, times characterized by long-term decreases in global temperature and biodiversity, in addition to the formation of the supercontinents. There is grammatical problems in the long sentence. Please revise.

Page 2: The later work devoted to the Cretaceous turn of geological evolution [2], however, showed that the galactic summers were associated with peaks of various igneous activities including both plume- and sub[1]duction/collision-related magmatism, while the galactic winters led to significant drops of any igneous activity. This sentence is too long to understand the meaning. Thus, please rephrase.

Page 2: The more recent works [4, 5] demonstrated a close relationship between plume- and subduction/collision-related activities in Northeast Asia for the Mesozoic time and, thus supported the latter supposition. This sentence is too long to understand the meaning. Thus, please rephrase. Besides, the latter is what? Authors should state clearly.

Page 2: ‘Below, we will test our previous ideas reviewing the Phanerozoic volcanic activity of Large Igneous Provinces (LIPs) focusing on the largest and best-studied provinces oc[1]curred on continents’ should be ‘Below, we will test our previous ideas by reviewing the Phanerozoic volcanic activities of Large Igneous Provinces (LIPs), especially the largest and best-studied provinces occurred on continents’.

Page 3: ‘The LIPs names are shown only for the provinces, which geochemical data are used in this study’ should be ‘The LIPs names are shown only for their provinces, whose geochemical data are used in this study’.

Page 5: ‘Note, however, that the Kalkarindji LIP, which occurred in the older galactic-summer time, may not be identified with the available data (Figure 2A)’ should be ‘Of note, the Kalkarindji LIP, which occurred in the older galactic-summer time, may not be identified with the available data (Figure 2A)’.

Page 9: In the conclusion section, please delete the reference [Nechaev, 2004].

Page 9: ‘…, that is indicated by peak abundances of ironstone ores’ should be ‘…, which is indicated by peak abundances of ironstone ores’.

Author Response

The manuscript entitled ‘Phanerozoic evolution of continental large igneous provinces: Implications for galactic seasonality’ by Nechaev et al., reviews the available data from the global Phanerozoic plume activities and geochemistry of their igneous rocks to find the evolutionary changes in intensity and geochemistry of mantle plumes. The topic is good for current climatic change. Although I am not an expert of the continental large igneous provinces and climatology, I try to do a careful reading the manuscript. I feel that the discussion section is a little too simple and suggest that authors should further discuss on the relationship between galactic seasonality and continental large igneous provinces in details.

Response: Thanks for this evaluation. We avoid to discuss many details in this review to present our general ideas clearer and easier to understand for a wide range of readers.

Besides, many grammatical problems exist in the text and language should be polished thoroughly by a native speaker. The followings are my comments on this manuscript. 

Page 1: ‘The materials subducted into the transitional and lower mantle highly influenced the plume mag[1][1]mas in the galactic-summer times and less significantly in the galactic spring and autumn’ should be ‘The materials subducted into the transitional and lower mantle, which highly influenced the plume mag[1][1]mas in the galactic-summer times and less significantly in the galactic spring and autumn’.

Page 1: ‘The least subduction-influenced probably were the Tarim and Emeishan deep plume magmas…’ should be ‘The least subduction-influenced probably was the Tarim and Emeishan deep plume magmas…’.

Page 1: ‘…, that is indicated by peak abundances of ironstone ores’ should be ‘…, which is indicated by peak abundances of ironstone ores’.

Page 1: ‘while that star was flying along its elliptical orbit’ should be ‘while stars were flying along its elliptical orbit’.

Page 1: Galactic winters occurred during the Vendian, Carboniferous-Permian and Late Cenozoic periods, times characterized by long-term decreases in global temperature and biodiversity, in addition to the formation of the supercontinents. There is grammatical problems in the long sentence. Please revise.

Page 2: The later work devoted to the Cretaceous turn of geological evolution [2], however, showed that the galactic summers were associated with peaks of various igneous activities including both plume- and sub[1][1]duction/collision-related magmatism, while the galactic winters led to significant drops of any igneous activity. This sentence is too long to understand the meaning. Thus, please rephrase.

Page 2: The more recent works [4, 5] demonstrated a close relationship between plume- and subduction/collision-related activities in Northeast Asia for the Mesozoic time and, thus supported the latter supposition. This sentence is too long to understand the meaning. Thus, please rephrase. Besides, the latter is what? Authors should state clearly.

Page 2: ‘Below, we will test our previous ideas reviewing the Phanerozoic volcanic activity of Large Igneous Provinces (LIPs) focusing on the largest and best-studied provinces oc[1][1]curred on continents’ should be ‘Below, we will test our previous ideas by reviewing the Phanerozoic volcanic activities of Large Igneous Provinces (LIPs), especially the largest and best-studied provinces occurred on continents’.

Page 3: ‘The LIPs names are shown only for the provinces, which geochemical data are used in this study’ should be ‘The LIPs names are shown only for their provinces, whose geochemical data are used in this study’.

Page 5: ‘Note, however, that the Kalkarindji LIP, which occurred in the older galactic-summer time, may not be identified with the available data (Figure 2A)’ should be ‘Of note, the Kalkarindji LIP, which occurred in the older galactic-summer time, may not be identified with the available data (Figure 2A)’.

Page 9: In the conclusion section, please delete the reference [Nechaev, 2004].

Page 9: ‘…, that is indicated by peak abundances of ironstone ores’ should be ‘…, which is indicated by peak abundances of ironstone ores’.

Response: The second author of this review, Prof. F. Lin Sutherland is a native English speaker. He has corrected and polished thoroughly the original text written by the first author of the manuscript. Nevertheless, all the corrections suggested by Reviewer 2 below have been considered and accepted in slightly modified form in the current MS revision. In addition, Prof. F. Lin Sutherland has made some other edits within the text that seemed needed for reviewers wishes for greater clarity. All these language editions are shown as light blue insertions in the revised manuscript.

Reviewer 3 Report

The manuscript entitled “Phanerozoic evolution of continental large igneous provinces: Implications for galactic seasonality” This study reviews the available data on the Phanerozoic plume activity (LIP’s size and frequency) and geochemistry of their igneous rocks. They conclude that the Phanerozoic plume activity including the LIP’s frequency and size were significantly higher in the Cambrian-Ordovician and, more distinctly, Jurassic-Cretaceous galactic-summer times, whereas while the galactic winters led to significant drops of any igneous activity. Furthermore, they believe that the least subduction-influenced probably were the Tarim and Emeishan deep plume magmas occurred in the mid-late Permian, during the galactic late winter-early spring subseason, and the Fe enrichment of clinopyroxenite, gabbro and associated Fe-Ti ores of these provinces might be, at least partly, caused by fluids ascending from the core-mantle boundary. I read it interestingly, and I believe that the idea is novel. However, the interpretation is too speculative, especially proxy for the mantle sources. They should discuss this issue in detail. It is noted that many authors have proposed that the Emeishan and Tarim LIP have been affected by subduction, and the formation of associated Fe-Ti ores of these provinces have been correlated to involvement of subducted oceanic crust as listed below:

Emeishan LIP:

1.       Hou, T., Zhang, Z.C., Ye, X.R., Encarnacion, J. and Reichow, M.K., 2011. Noble gas isotopic systematics of Fe-Ti-V oxide ore-related mafic-ultramafic layered intrusions in the Panxi area, China, The role of recycled oceanic crust in their petrogenesis. Geochimica et Cosmochimica Acta 75, 6727-6741.

2.       Hou, T., Zhang, Z.C., Encarnacion, J., Santosh, M. and Sun, Y.L., 2013. The role recycled oceanic crust in magmatism and metallogenesis, Os-Sr-Nd isotpes, U-Pb geochronology and geochemistry of picritic dykes in the Panzhihua giant Fe-Ti oxide deposit, central Emeishan large igneous province. Contributions to Mineralogy and Petrology, 165, 805-822.

3.       Bai, Z.J., Zhong, H., Li, C.S., Zhu, W.G., He, D.F., Qi, L., 2014. Contrasting parental magma compositions for the Hongge and Panzhihua magmatic Fe-Ti-V oxide deposits, Emeishan large igneous province, SW China. Economic Geology, 109, 1763-1785.

4.       Yu, S.Y., Shen, N.P., Song, X.Y., Ripley, E.M., Li, C.S., Chen, L.M., 2017. An integrated chemical and oxygen isotopic study of primitive olivine grains in picrites from the Emeishan Large Igneous Province, SW China: Evidence for oxygen isotope heterogeneity in mantle sources. Geochimica et Cosmochimica Acta 215 , pp.263-276

5.       Zhu, J., Zhang, Z.C., Reichow, M. K., Li, H.B., Cai, W.C., & Pan, R.H., 2018. Weak vertical surface movement caused by the ascent of the Emeishan mantle anomaly. Journal of Geophysical Research: Solid Earth, 123, 1018-1034.

Tarim LIP

1.       Zhang, D.Y., Zhang, Z.C., Santosh M., Cheng, Z.G., Huang, H. and Kang, J.L., 2013. Perovskite and baddeleyite from kimberlitic intrusions in the Tarim large igneous province signal the onset of an end-Carboniferous mantle plume. Earth and Planetary Science Letters, 361(1), 238-248.

2.       Cheng, Z.G., Zhang, Z.C., Xie, Q.H., Hou, T., Ke, S., 2018. Subducted slab-plume interaction traced by magnesium isotopes in the northern margin of the Tarim Large Igneous Province. Earth and Planetary Science Letters, 489, 100-110

3.       Cheng, Z.G., Zhang, Z.C., Hou, T., Santosh, M., Chen, L.L., Ke, S. and Xu, L.J., 2017.  Decoupling of Mg-C and Sr-Nd-O isotopes traces the role of recycled carbon in magnesiocarbonatites from the Tarim Large Igneous Province. Geochimica et Cosmochimica Acta, 202, 159-178

4.       Cheng, Z.G, Zhang, Z.C., Wang, Z.C., Wang, F.Y., Mao, Q., Xu, L.J., Ke, S., Yu, H.M., Santosh, M., 2020. Petrogenesis of transitional large igneous province: Insights from bimodal volcanic suite in the Tarim large igneous province. Journal of Geophysical Research: Solid Earth, 125, e2019JB018382.

5.       Zhu, S.Z., Huang, X.L., Yang, F., He, P.L., 2021. Petrology and geochemistry of early Permian mafic-ultramafic rocks in the Wajilitag area of the southwestern Tarim Large Igneous Province: Insights into Fe-rich magma of mantle plume activity. Lithos, 398, 106355.

6.       Wang, C.H., Zhang, Z.C., Giuliani, A., Cai, R.H., Cheng, Z.G., Liu, J.G., 2022. New insights into the mantle source of a large igneous province from highly siderophile element and Sr-Nd-Os isotope compositions of carbonate-rich ultramafic lamprophyres. Geochimica et Cosmochimica Acta, 326, 77-96.

In addition, the authors should define what are the “spring”, “summer”, “autumn” and winter”. I believe that most readers would be confused by these terms.

Author Response

The manuscript entitled “Phanerozoic evolution of continental large igneous provinces: Implications for galactic seasonality” This study reviews the available data on the Phanerozoic plume activity (LIP’s size and frequency) and geochemistry of their igneous rocks. They conclude that the Phanerozoic plume activity including the LIP’s frequency and size were significantly higher in the Cambrian-Ordovician and, more distinctly, Jurassic-Cretaceous galactic-summer times, whereas while the galactic winters led to significant drops of any igneous activity. Furthermore, they believe that the least subduction-influenced probably were the Tarim and Emeishan deep plume magmas occurred in the mid-late Permian, during the galactic late winter-early spring subseason, and the Fe enrichment of clinopyroxenite, gabbro and associated Fe-Ti ores of these provinces might be, at least partly, caused by fluids ascending from the core-mantle boundary. I read it interestingly, and I believe that the idea is novel. However, the interpretation is too speculative, especially proxy for the mantle sources. They should discuss this issue in detail. It is noted that many authors have proposed that the Emeishan and Tarim LIP have been affected by subduction, and the formation of associated Fe-Ti ores of these provinces have been correlated to involvement of subducted oceanic crust as listed below:

Emeishan LIP:

  1. Hou, T., Zhang, Z.C., Ye, X.R., Encarnacion, J. and Reichow, M.K., 2011. Noble gas isotopic systematics of Fe-Ti-V oxide ore-related mafic-ultramafic layered intrusions in the Panxi area, China, The role of recycled oceanic crust in their petrogenesis. Geochimica et Cosmochimica Acta 75, 6727-6741.
  2. Hou, T., Zhang, Z.C., Encarnacion, J., Santosh, M. and Sun, Y.L., 2013. The role recycled oceanic crust in magmatism and metallogenesis, Os-Sr-Nd isotpes, U-Pb geochronology and geochemistry of picritic dykes in the Panzhihua giant Fe-Ti oxide deposit, central Emeishan large igneous province. Contributions to Mineralogy and Petrology, 165, 805-822.
  3. Bai, Z.J., Zhong, H., Li, C.S., Zhu, W.G., He, D.F., Qi, L., 2014. Contrasting parental magma compositions for the Hongge and Panzhihua magmatic Fe-Ti-V oxide deposits, Emeishan large igneous province, SW China. Economic Geology, 109, 1763-1785.
  4. Yu, S.Y., Shen, N.P., Song, X.Y., Ripley, E.M., Li, C.S., Chen, L.M., 2017. An integrated chemical and oxygen isotopic study of primitive olivine grains in picrites from the Emeishan Large Igneous Province, SW China: Evidence for oxygen isotope heterogeneity in mantle sources. Geochimica et Cosmochimica Acta 215 , pp.263-276
  5. Zhu, J., Zhang, Z.C., Reichow, M. K., Li, H.B., Cai, W.C., & Pan, R.H., 2018. Weak vertical surface movement caused by the ascent of the Emeishan mantle anomaly. Journal of Geophysical Research: Solid Earth, 123, 1018-1034.

Tarim LIP

  1. Zhang, D.Y., Zhang, Z.C., Santosh M., Cheng, Z.G., Huang, H. and Kang, J.L., 2013. Perovskite and baddeleyite from kimberlitic intrusions in the Tarim large igneous province signal the onset of an end-Carboniferous mantle plume. Earth and Planetary Science Letters, 361(1), 238-248.
  2. Cheng, Z.G., Zhang, Z.C., Xie, Q.H., Hou, T., Ke, S., 2018. Subducted slab-plume interaction traced by magnesium isotopes in the northern margin of the Tarim Large Igneous Province. Earth and Planetary Science Letters, 489, 100-110
  3. Cheng, Z.G., Zhang, Z.C., Hou, T., Santosh, M., Chen, L.L., Ke, S. and Xu, L.J., 2017.  Decoupling of Mg-C and Sr-Nd-O isotopes traces the role of recycled carbon in magnesiocarbonatites from the Tarim Large Igneous Province. Geochimica et Cosmochimica Acta, 202, 159-178
  4. Cheng, Z.G, Zhang, Z.C., Wang, Z.C., Wang, F.Y., Mao, Q., Xu, L.J., Ke, S., Yu, H.M., Santosh, M., 2020. Petrogenesis of transitional large igneous province: Insights from bimodal volcanic suite in the Tarim large igneous province. Journal of Geophysical Research: Solid Earth, 125, e2019JB018382.
  5. Zhu, S.Z., Huang, X.L., Yang, F., He, P.L., 2021. Petrology and geochemistry of early Permian mafic-ultramafic rocks in the Wajilitag area of the southwestern Tarim Large Igneous Province: Insights into Fe-rich magma of mantle plume activity. Lithos, 398, 106355.
  6. Wang, C.H., Zhang, Z.C., Giuliani, A., Cai, R.H., Cheng, Z.G., Liu, J.G., 2022. New insights into the mantle source of a large igneous province from highly siderophile element and Sr-Nd-Os isotope compositions of carbonate-rich ultramafic lamprophyres. Geochimica et Cosmochimica Acta, 326, 77-96.

Response: Thanks for your evaluation of our ideas as interesting. Of course, our hypothesis may be considered speculative like any hypothesis before it becomes a theory. Many thanks also for the suggested papers, which undoubtedly evidence “that the Emeishan and Tarim LIP have been affected by subduction, and the formation of associated Fe-Ti ores of these provinces have been correlated to involvement of subducted oceanic crust”. However, we do not completely deny the effect of subduction on the Tarim and Emeishan magmas. Moreover, these studies do not state the temporal correlation between subduction and plume processes. For example, Hou et al. (2011), who revealed low 3He/4He and 40Ar/36Ar ratios of olivine and clinopyroxene grains separated from the Fe–Ti–V oxide ore-bearing intrusions in the Emeishan LIP, concluded that they are due to “subduction-related fluids and melts that had been stored in the lithospheric mantle for long periods. Considering the regional geologic history, such addition can be attributed to the paleo subduction that occurred along the western margin of the Yangtze Block during the Neoproterozoic”. Thus, the results of many studies referenced by Reviewer 3 do not contradict to our suggestion that the Tarim and Emeishan deep plume magmas were the least influenced by contemporary subduction. The corresponding additions to the main text (the penultimate paragraph of Section 3.3 on Page 9) and References [36-45] have been added.

In addition, the authors should define what are the “spring”, “summer”, “autumn” and winter”. I believe that most readers would be confused by these terms.

Response: The galactic seasons are defined in the first paragraph of Section 1 and illustrated in Fig. 1. More details may be found in our previous publications [1,2] that are freely available.

Round 2

Reviewer 2 Report

I have no question and agree to publish.

Reviewer 3 Report

I have no other comments or suggestions. The authors have addressed all my previous comments. The revised version sounds reasonable, and can be accepted for publication.